# Vitamin D and Its Receptor from a Structural Perspective

**DOI:** 10.3390/nu14142847

**Published:** 2022-07-12

**Authors:** Natacha Rochel

**Affiliations:** 1Integrated Structural Biology Department, Institut de Génétique et de Biologie Moléculaire et Cellulaire, 67404 Illkirch, France; rochel@igbmc.fr; 2Centre National de la Recherche Scientifique, UMR7104, 67404 Illkirch, France; 3Institut National de la Santé et de la Recherche Médicale (INSERM), U1258, 67404 Illkirch, France; 4Université de Strasbourg, 67404 Illkirch, France

**Keywords:** vitamin D, vitamin D receptor, 3D structure, structural analysis, molecular recognition, protein–ligand interactions, coregulators

## Abstract

The activities of 1α,25-dihydroxyvitamin D3, 1,25D_3_, are mediated via its binding to the vitamin D receptor (VDR), a ligand-dependent transcription factor that belongs to the nuclear receptor superfamily. Numerous studies have demonstrated the important role of 1,25D_3_ and VDR signaling in various biological processes and associated pathologies. A wealth of information about ligand recognition and mechanism of action by structural analysis of the VDR complexes is also available. The methods used in these structural studies were mainly X-ray crystallography complemented by NMR, cryo-electron microscopy and structural mass spectrometry. This review aims to provide an overview of the current knowledge of VDR structures and also to explore the recent progress in understanding the complex mechanism of action of 1,25D_3_ from a structural perspective.

## 1. Introduction

Upon sun exposure, the secosteroid prohormone vitamin D is two-step hydroxylated in the liver and the kidney into the biologically hormonal form, 1α,25-dihydroxyvitamin D3 (1,25D_3_) [1]. Most effects of 1,25D_3_ are mediated via its binding to the vitamin D receptor (VDR, also termed NR1I1), a ligand-dependent transcription factor (TF) that belongs to the nuclear receptor (NR) superfamily. VDR was cloned and sequenced in mammalian and avian species in the late 1980′s [2,3,4,5]. VDR acts as a heterodimer with one of the three retinoid X receptor (RXR) isotypes (RXRα, NR2B1; RXRβ, NR2B2; and RXRγ, NR2B3). VDR and its ligand control calcium metabolism, cell growth, differentiation, anti-proliferation, apoptosis and adaptive/innate immune responses [6,7,8]. VDR is expressed in the different tissues of the body and many of these tissues were not originally considered target tissues for 1,25D_3_. Conversion of 25D_3_ to 1,25D_3_ also occurs in many non-renal tissues and cells, including the skin, parathyroid glands, bone cells, both cardiovascular and immune cells, and many others [9,10]. The discovery of these autocrine/paracrine activities of 1,25D_3_ has markedly increased our appreciation of the wide effects of 1,25D_3_. An additional layer of complexity came with the identification of alternative vitamin D pathways and the major enzymes involved that produce other natural vitamin D metabolites, with some of them showing potent VDR activation [11,12]. Deregulation of VDR function may lead to severe diseases, such as cancers, psoriasis, rickets, renal osteodystrophy, and autoimmunity disorders (multiple sclerosis, rheumatoid arthritis, inflammatory bowel diseases, type I diabetes) [13]. Despite the large number of potential applications, clinical use of the native hormone 1,25D_3_ is limited by calcification of soft tissues (hypercalcemia). However, synthetic analogs have been developed and some of the highly active and non-calcemic VDR ligands have found clinical applications in the standard topical treatment of psoriasis, secondary hyperparathyroidism or osteoporosis [14,15].

In this review, we discuss the general mechanism of action of VDR and what we have learned from structural studies of VDR complexes from isolated domains to full length proteins focusing on 1,25D_3_ action. Recent studies that aimed to describe how coregulators interact with VDR will also be discussed. These studies have significantly advanced our understanding on the molecular mechanism by which 1,25D_3_ mediates transcription regulation. However, important questions remain regarding the basic mechanism of the cell-specific action of ligands and possible cross-talks with other NRs and TFs. Finally, the ongoing effort to characterize the structures, dynamics and relationships with the function of large VDR coregulatory complexes will be discussed. 

## 2. Structure and Mechanism of Activation of the Vitamin D Receptor

VDR shares two main features with other members of the NR superfamily, principal mechanism of action and structural organization. Simplified regulation of target gene expression by VDR can be presented as follows (Figure 1): VDR forms a heterodimer with RXR and binds to specific DNA sequences of controlled genes called VDR response element (VDRE). These VDREs are often localized thousands of base pairs from the coding region of the regulated gene [16,17]. Typically, in the absence of ligands or in the presence of antagonists, corepressors with histone deacetylase activity are recruited to VDR bound to their target genes, while binding of agonist ligands induces a change in the structure of the NR that allows interaction with coactivators [8,16]. Recruitment of coactivators with enzymatic activities, such as histone acetyl-transferases (HAT), prepares target gene promoters through decondensation of the chromatin. HATs can further be replaced by the mediator complex that provides a link with the basal transcriptional machinery.

The second feature shared by VDR and other NRs is its structural organization. VDR is a molecule of approximately 50–60 kDa, depending on species. The human VDR has two potential start sites with a common polymorphism (Fok 1) that alters the first ATG start site to ACG, leading to a VDR that is three amino acids shorter (424 AA vs. 427 AA), a polymorphism correlated with reduced bone density [18]. The VDR has a modular organization (Figure 2A) that consists of a short variable and flexible N-terminal domain, a highly conserved DNA-binding domain (DBD), a conserved ligand-binding domain (LBD) and a hinge region connecting the DBD to LBD. In addition, VDR has a unique feature among NRs, with its long insertion region within the LBD that is in a disordered state [19]. Since the first crystal structures of VDR LBD in 2000 [20] and of VDR DBD in 2002 [21] (Figure 2B,C), numerous studies have investigated the isolated domains of VDR, mainly by X-ray crystallography complemented by NMR and hydrogen-deuterium exchange coupled with mass spectrometry (HDX-MS).

## 3. DNA Binding

The DBD, the most conserved domain in VDR from different species and among the NRs, is comprised of two zinc fingers (Figure 2B). The first zinc finger is important for specific DNA binding to the VDREs, while the second one is involved in heterodimer interaction. Steric constraints of the VDR–RXR complex determine the optimal heterodimer binding site within VDRE as a direct repeat of the sequence RGKTSA (R = A or G, K = G or T, S = C or G), separated by three nucleotides (DR3) [22,23,24]. RXR binds to the upstream half site, while VDR binds to the downstream site. The crystal structure of the VDR DBD homodimer on DR3 [21] has revealed that the key interactions between the VDR DBD and the DNA, including four conserved residues in the recognition helix, Glu42, Lys45, Arg49 and Arg50, make sequence-specific base contacts in the major groove of the half-site. ChIP-seq studies in various cell types (see references [25,26,27,28,29,30,31] among others) have confirmed that the DR3 are the most enriched motifs upon ligand treatment but that represent only 10–20% of all VDR binding sites and most of VDREs spread over the whole genome [17]. A prerequisite for VDR DNA binding is the accessibility of the binding site through the action of pioneer factors and coactivators to open the chromatin and to modify chromatin topology [17,24,25].

## 4. Ligand Binding

NR is a highly dynamic scaffold protein and VDR LBD (Figure 2C) in a similar way to other NRs, is dynamic and only stabilized into a fixed conformation upon ligand binding. The dynamics of ligand binding process by VDR has been investigated by NMR [32] and HDX-MS [33,34,35]. These studies revealed that the entire C-terminal of VDR LBD in its apo state is very dynamic, with 80% of amide hydrogen exchange. The region forming the ligand binding pocket (LBP) (Figure 2C) also showed a high exchange rate, while the central layer of the α-helical sandwich appeared to be protected. Binding of 1,25D_3_ has been shown to lead to significant protection from hydrogen amide exchange, not only for the LBP but also in regions remote from the LBP.

Detailed information on the binding mode of 1,25D_3_ has been obtained by the elucidation of the crystal structure by X-ray crystallography of its complex with the human VDR LBD [20]. For the crystallization of the hVDR–1,25D_3_ complex, a truncated form of the hVDR LBD was used that lacks the insertion domain (Figure 2A). This region is characterized by poor sequence conservation between VDR family members, is predicted to be disordered and does not play a major role in receptor selectivity for 1,25D_3_ [19,36]. The general fold of VDR LBD (PDB IDs: 1DB1 and 7QPP) consists of a three-layered α-helical sandwich composed of twelve helices (H1 to H12), three two-turn helices (H3n, H4n and Hx) and a three-stranded β-sheet (Figure 3A). The LBP is surrounded by helices H2, H3, H5, H6, H7, H10 an H12. The residues of each of β-sheet strands also form contacts with the ligand.

The ligand occupies 56% of the volume of the LBP (697 Å^3^) with some water molecules near the position 2 of the A-ring of 1,25D_3_ (Figure 3B). The ligand adopts a chair B conformation with the 19-methylene “up” and the 1α-OH and 3β-OH groups in equatorial and axial orientations, respectively, while the aliphatic chain at position 17 of the D-ring adopts an extended conformation. The ligand is anchored in the LBP through three pair of hydrogen bonds formed between three hydroxyl groups of the ligand and polar residues: 1-OH group with Ser237 (H3) and Arg274 (H5), 3-OH group with Ser278 (H5) and Tyr143 (loop H1-H2) and 25-OH group with His305 (loop H6-H7) and His397 (H11) (Figure 3B). In addition, the ligand interacts with the hydrophobic residues lining the LBP (Figure 3C).

The LBD also contains the regions necessary for heterodimerization to RXR, comprising H9 and H10 and the loop 8–9, and for coactivator interaction through the activation function AF-2 formed by H3, H4 and H12. Helix 12 closed the LBP and is stabilized by two interactions with the ligand. Helix 12 is also stabilized by several hydrophobic contacts with residues of H3, H5 and H11 and two polar interactions with residues of H3 and H4. Some of these residues contact the ligand, thus indicating an additional indirect ligand-control of the position of helix H12.

The structures of 1,25D_3_ in the complex with VDR of other species were also described for *Rattus norvegicus* (rVDR) [37], *Danio rerio* (zebrafish, zVDRα) [38] and *Petromyzon marinus* (sea lamprey (l)) [39], the most basal vertebrate showing the most divergent VDR sequence. In the case of the rVDR LBD complex (PDB ID: 1RK3), the same truncation of the large insertion region connecting helices H1 to H3 as for hVDR was applied. For the zVDR (PDB ID: 2HC4) and lVDR (PDB ID: 7QPI), the wild-type LBD was used. The binding mode of 1,25D_3_ to VDR LBP is similar in all VDR structures, indicating a conserved ligand selectivity of VDRs across vertebrate species. While the differences are small between the structures of h, r and z VDR LBDs and primarily involve the loops, some significant differences are observed for lVDR around the linker regions between H11-H12 and H9-H10. These differences explain the weaker AF-2 stabilization and weaker RXR dimerization [39], and consequently the lower efficacy to activate lVDR compared to higher vertebrate VDRs [40,41]. Interpretation of the VDR structures and VDR–ligand interaction differences in a phylogenetic context allow for significant progress towards understanding the molecular activities. Indeed, structural and sequence co-evolution analysis allow to identify the conserved residues to maintain their functional integrity, and pairs of amino acids that coevolved to accommodate novel functionalities. Coevolving residues are located in H9 and in the insertion domain [39], accounting for the increased sensitivity to RXR and coregulators during evolution and leading to the increase in transactivation responses to 1,25D_3_ and VDR to be fully activated by 1,25D_3_ and to respond to lithocholic acid in higher vertebrates, facilitating novel functions.

**Figure 3 nutrients-14-02847-f003:**
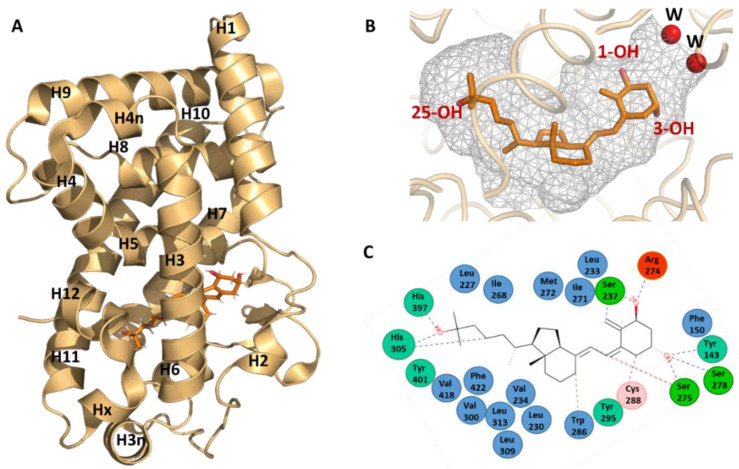
1,25D_3_ recognition by VDR. (**A**) Overall structure of the hVDR LBD. The hVDR LBD bound to 1,25D_3_ is composed of 15 helices and 1 β-sheet (PDB ID: 1DB1 and 7QPP) [20,39]. (**B**) Conformation of 1,25D_3_ in the VDR LBP shown as a grey surface. (**C**) Interaction map of 1,25D_3_ in the LBP of hVDR.

This structural knowledge has been pivotal in the comprehension of VDR action and to understand the impact of mutations found in hereditary vitamin D-resistant rickets (HVDRR) patients showing a phenotype connected with softening and weakening of the bones [42,43]. The alterations in the VDR gene are caused by mutations that result in suboptimal gene regulatory responses, despite the presence of 1,25D_3_ in the body. To date, 12 missense mutations have been identified in hVDR DBD, all associated with alopecia [43]. In the LBD, 24 amino acids of the hVDR were found mutated in HVDRR patients, some associated with alopecia [43]. The mutations result in alterations of VDR functions as ligand- or DNA-binding, heterodimerization with RXR or coregulator-binding. However, the overall effect of the mutation may result from the combination effects of all four functions at the same time. Among those mutations, the mutations involved residues forming hydrogen bonds with the hydroxyl groups of 1,25D_3_, including Arg274Leu/His [44,45,46], His305Gln [47] and His397Pro [48], the last one being associated with very severe HVDRR. The discussion of the structural implications of these missense mutations were presented in the studies [49,50].

In addition, the crystal structures of VDR LBD have provided significant information for the characterization and the design of more specific analogs. A large part of the VDR LBP remains unoccupied when bound by 1,25D_3_ (Figure 3B), providing additional space for the fitting of the modified moieties of the hormone and some analogs induce significant conformational changes, enlarging the original LBP (review in [15,51,52]). Not only secosteroidal analogs, but also other VDR ligands with non secosteroidal structures, such as lithocholic acid derivatives and mimics of 1,25D_3,_ have been reported. Hundreds of analogs in the complexes with VDR LBD from human, rat or zebrafish species were crystalized. The binding mode of some of these analogs to VDR LBP and their mechanism of action were discussed in [15,51,52]. In addition to the development of molecules targeting the VDR LBP and acting as agonist or antagonist, small molecules that target VDR–coregulator binding and act as protein-protein inhibitors are being developed [53]. Atomic-level understanding allows for the development of compounds with more original chemical structures and more specific action. However, due to the complexity of VDR signaling, these compounds remain largely unsuccessful in reaching therapeutic applications thus far.

## 5. Structure of Full-Length VDR Complex

Until now, only the VDR LBD monomer has been crystallized. The lower affinity of VDR for RXR compared to other NRs, such as PPAR or RAR [39,54], may explain the difficulty to crystallize the VDR–RXR LBDs heterodimer. DNA binding stabilizes the VDR heterodimer and 1,25D_3_ is required for high affinity binding and activation. On other hand, RXR ligand, 9-cis retinoic acid, has been shown to either inhibit [55] or activate [56,57] -1,25D_3_ stimulation of gene transcription. In the absence of a full-length heterodimer crystal structure, solution methods using small angle X-ray scattering (SAXS) [58], cryo-electron microscopy (cryoEM) [59] and HDX-MS [60] have provided information on the full length VDR–RXR–DNA complex. These methods render it possible to visualize how the DBD and the LBD/heterodimerization domains are arranged relative to one another and how their binding to ligand, DNA, and coactivators influence one another.

The LBD and DBD domains in the full length structure is structurally conserved, compared to those of previously solved individual domain structures. The solution structures of VDR–RXR show that DBDs and LBDs are separated and positioned asymmetrically (Figure 4). The relative position of the domains and the observed asymmetry of the overall architecture both point to the essential role played by the hinge domains in establishing and maintaining the integrity of the functional structures. While the RXR hinge does not have a well-defined structure, the hinge domain of VDR forms an α-helix that stabilizes the whole complex, thus facilitating the positioning of the LBD and the surface to be accessible by the coregulators [58,59]. The 3D structures of several NRs truncated by their NTDs and in complex with DNA have now been obtained by X-ray crystallography or cryoEM for PPAR–RXR [61], HNF4 [62], LXR–RXR [63], RAR–RXR [64], EcR–USP [65] and AR [66]. These studies suggest that NRs are rather flexible macromolecules, adapting several conformations. In the VDR–RXR complex, even with the relatively separated positioning, there is evidence of long-range allosteric connections between the VDR LBD, DBD and DNA [60]. HDX-MS was used to understand the conformational plasticity and allosteric/dynamic communications in VDR complexes and has revealed cooperative effects between the VDR DBD and LBD to fine tune transcriptional regulation by the ligands and the DNA [67]. Upon 1,25D_3_ binding to full VDR–RXR, the differential HDX experiment has revealed a profile very similar to the binding to VDR LBD alone with a stabilization of VDR H12. Interestingly, an increase in solvent exchange in the DBD of VDR was also observed upon ligand binding, indicating that the ligand impact on the DBD conformation. Upon ligand binding, a stabilization of the heterodimer interface was also observed. The HDX profile of 1,25D_3_ binding in the presence of RXR ligand was similar to that of 1,25D_3_ alone. Allosteric communication is ligand dependent and bidirectional. DNA binding modulates both DBD–DBD and LBD–LBD interactions and influences coactivator recognition and binding. Conformational dynamics indicate that the binding of VDR–RXR to DNA results in significant alterations in the conformation of the LBD within the region important for interactions with coactivators, VDR H12 and RXR H3, and dimer interface H10–H11. Difference in DNA sequences modulate the receptor dynamics in remote regions, such as the coactivator binding surfaces [60,67]. A similar effect was observed for the RAR–RXR–DNA complex [68].

## 6. VDR–Coregulatory Complexes

The successful regulation of transcription by NRs requires the recruitment of coregulators to genomic loci, an event that directly affects the transcriptional rate. Hundreds of NR coregulators have been reported and include coactivators and corepressors [69]. For VDR, the mechanism of coactivation is now well understood, while VDR corepression is less studied. The major VDR coactivators are the NCoAs and mediator complexes [70,71]. The NCoAs (NCoA1, NCoA2 and NCoA3) recruit secondary coactivators CBP/p300 and p/CAF that have histone acetyl transferase activity and interact with VDR in a 1,25D_3_-dependent manner [71]. Another important VDR coactivator is MED1, a subunit of the mediator that links the TFs to the transcription machinery [72].

Most of the coactivators that can be recruited to the NRs in a ligand-dependent way contain the conserved LXXLL motif or NR box [73]. An analogous sequence motif (LXXH/IIXXXI/L) was identified in corepressors [74,75,76]. NR boxes are often located within intrinsically disordered regions of the coactivators and corepressors. NCoAs contain several domains separated by long disordered regions, the N-terminal part contains a highly conserved basic helix-loop-helix (HLH) and a signaling PAS (Per/Arnt/Sim) domain that mediates protein–protein interactions. The NRs interact with NCoAs via the receptor interaction domain (RID) that contains three LXXLL motifs. Each domain in NCoAs specializes in recruiting various TFs or other coregulators of transcription, including protein-modifying enzymes and chromatin remodelers. VDR has been shown to preferentially interact with the second and third LXXLL motifs of the NCoA RID [77]. MED1 contains two NR boxes in its central RID that differentially bind to NRs [78]. The second motif has been shown to preferentially bind to VDR [77]. These coregulators are not specific to VDR, but interact with a large number of other NRs and transcription factors. The mediator complex can be found in large numbers of loci and are known to be involved as super enhancers [79]. In addition, other domains of MED1 outside the RID has been shown to interact with NRs [80].

Multiple biochemical and structural studies have mapped the interaction regions of NRs with the coactivator NR box [73,81,82,83] and numerous X-ray structures of NR LBDs with bound coregulator LXXLL peptides are nowadays deposited in the PDB. Analogous recognition region in the corepressors, CoR-NR box, binds in the same hydrophobic groove on LBDs. Therefore, the recruitment of coactivators and corepressors is mutually exclusive. The intrinsically disordered properties of the coregulators that are important to adapt and interact with many different transcription factors also limit their studies by X-ray crystallography or cryoEM.

MED1 NR2 as well as NCoA1 NR2, NCoA2 NR2 and NR3 were crystallized with liganded VDR LBD [37,39,84,85], revealing the binding mode of the LXXLL motifs to the VDR LBD. The LXXLL peptide formed a short a-helix that binds to a surface formed by helices H3, H4 and H12 of the LBD. The interaction of MED1 NR2 that buries about 507 Å^2^ of the receptor’s surface involves the three leucines that are buried within the pocket and surrounded by hydrophobic residues. The peptide is additionally locked through the formation of hydrogen bonds between conserved lysine residue in H3 and a glutamate residue in H12 that define a charge clamp.

Structures of coregulatory complexes with large fragment or full-length coactivators have been less studied due to their dynamic conformations. Several biophysical and solution structural studies in solution have been performed for some NRs bound to coactivator RIDs and recent developments in cryoEM have started to provide structural information on NR–coactivator complexes, including Erα-NcoA3-p300 [86] and AR-NcoA3-p300 [87]. However, NR–coregulator complexes structures have not reached atomic resolution yet, due to the conformational dynamics of the transcriptional complexes. For VDR, mass spectrometry structural methods combined with biophysical and structural methods have provided important details of VDR complex assembly and conformational plasticity and allosteric/dynamic communications within the complexes [60,67,88].

Differential HDX-MS has been used to study the interaction of VDR–RXR with the NCoA1 RID that contains the three LXXLL motifs [60,67]. As expected in the absence of both ligands, no coactivator interaction was observed and the separate addition of the 1,25D_3_ or RXR ligand, 9-cis retinoic acid, allow the coactivator to interact in a ligand-specific manner and independently. In the presence of 1,25D_3,_ only RXR AF-2 within the full VDR–RXR complex was insensitive to the binding of NCoA1 RID, indicating that it is primarily associated to VDR AF-2. However, RXR and its ligand modulate the interaction when VDR is liganded and NCoA1 RID binding stabilizes not only VDR AF-2 but also RXR H3 and H10-H11 [60]. Helices 3 and 4 of VDR that are part of the coactivator binding cleft cannot be further stabilized, since they achieve maximal stabilization upon 1,25D_3_ binding. For RXR, the loop between helices 10 and 11 is important in the formation of the hydrophobic groove facilitating coactivator binding. In addition, DNA binding modulates the interaction of NCoA1 with the heterodimer. Mutations of each LXXLL motifs and luciferase transactivation assays suggested that the third motif was associated to VDR and the first one to RXR [60]. Mutations on residues of VDR and RXR involved in the coactivator charge clamp indicate that the coactivator binding surface of each receptor is important for NCoA1 RID (Figure 5A). These data are in contrast with the SAXS data showing that MED1 RID mainly interacts with RXR’s heterodimeric partner [58]. Importantly, to fully understand the binding of MED1 to VDR–RXR, it was necessary to use a larger fragment of MED1 that not only contains the RID but also the N-terminal domain (50–660) [89]. Significant differences could be observed in comparison to the NCoA1 RID complex [89] by integrative structural methods combining SAXS, NMR and structural mass spectrometry. Differential HDX-MS and crosslink mass spectrometry confirmed that VDR interaction with MED1 motif NR2 is driving the complex formation but also demonstrated that other VDR–RXR regions outside the VDR AF-2, as well as MED1 regions other than RID, modulate the association and form an extended interaction surface (Figure 5B). Both LXXLL motifs of MED1 were perturbed upon formation of the complex with VDR–RXR, suggesting that while the NR1 box is not accommodated within the classical coactivator binding site, it could be either interacting with an alternative site of the receptors or stabilized allosterically. In addition to RID, the structured N-terminal domain of MED1 is also affected upon binding to VDR–RXR and is likely interacting with both VDR and RXR LBDs; in particular, the MED1 region 243–255 is largely stabilized in the complex with the receptor heterodimer. Crosslink MS also confirmed that this MED1 region is in physical proximity to the RXR. Among other novel MED1-interacting regions within the VDR–RXR heterodimer is the flexible insertion domain in the VDR LBD located between H1 and H3. Conformational changes upon interaction with MED1 was observed by NMR. This effect was not observed in the HDX-MS experiment; however, the observed difference could be attributed to the different temporal resolution of the two methods.

Differences in the binding modes could serve as molecular determinants of how the NRs discriminate between the coactivators. These studies were performed on large domains of coactivators but future studies should be carried out on the heterodimer complex with full length coactivators, as other domains, as shown for the N-ter domain of MED1, may modulate interactions directly or allosterically.

## 7. Conclusions and Perspectives

Our understanding of the molecular and structural insights for 1,25D_3_ action via its master regulator VDR has continuously advanced in the last twenty years, through the elucidation of the atomic structures of VDR DBD and LBD in complex with 1,25D_3_ and analogs. In addition, few structural studies on full length DNA bound VDR complexes have provided information on the allosteric effects driven by other domains and other effectors as DNA and coregulators. Important questions remain regarding the basic mechanisms of cell-specific action of ligands and possible cross-talks with other NRs and TFs. A deeper understanding of the interactions of VDR with specific coregulators will also be essential to better understand VDR action and may likely impact the future of drug development. However, full-length coregulators have large unstructured regions that remain a major hurdle for their structural characterization. CryoEM has started to provide structural information and will be essential to understand the structural dynamics of coregulatory complexes. The new structural biology tools, such as structural mass spectrometry, single molecule cryoEM with new processing tools and 3D classification, and cryo-tomography, will ultimately provide details on the structures of VDR complexes with coregulator complexes in a physiological environment.

## Figures and Tables

**Figure 1 nutrients-14-02847-f001:**
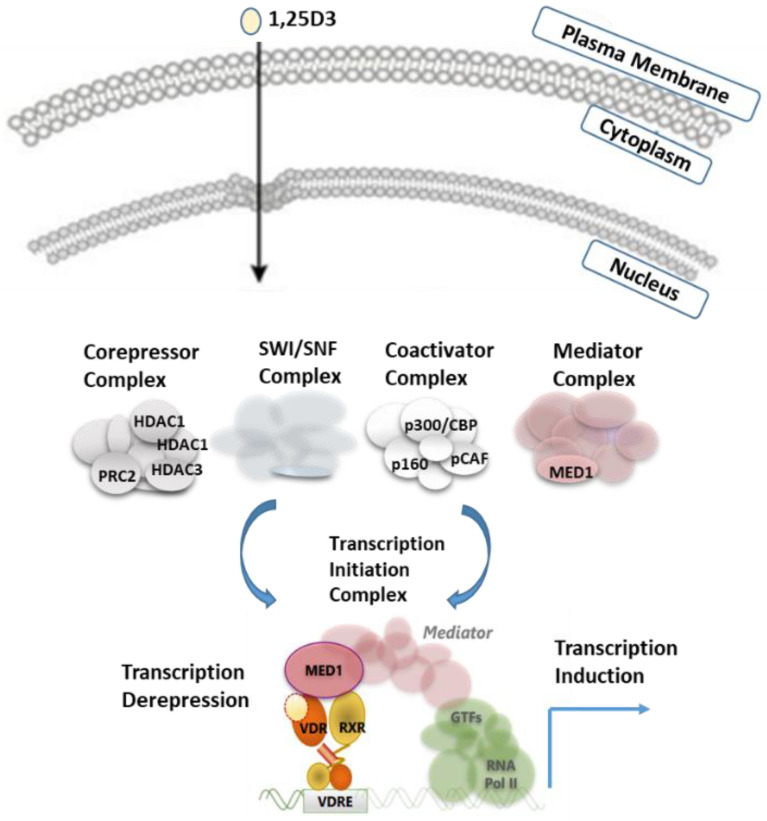
Schematic model for VDR regulation. Upon 1,25D_3_ binding to VDR, VDR translocates into the nucleus, binds as a heterodimer with RXR to DNA and interacts with various coregulators, leading to the activation of transcription or relief of constitutive repression.

**Figure 2 nutrients-14-02847-f002:**
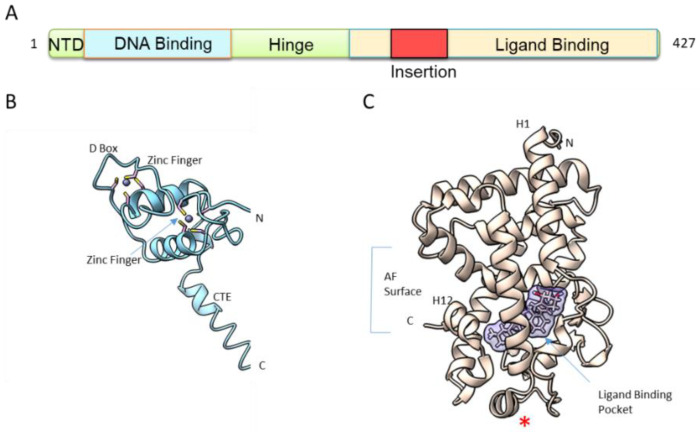
VDR structure. (**A**) Modular organization of VDR. NTD: N-terminal domain. DBD: DNA binding domain. LBD: ligand binding domain. (**B**) Overall structure of hVDR DBD monomer (PDB ID: 1KB4 [21]). Zn atoms are represented by spheres. CTE: C-terminal extension. (**C**) Overall structure of the hVDR LBD bound to 1,25D_3_ (PDB ID: 1DB1 [20]). LBP: ligand binding pocket. AF: activation function. The red star corresponds to the position of the truncation of the insertion domain.

**Figure 4 nutrients-14-02847-f004:**
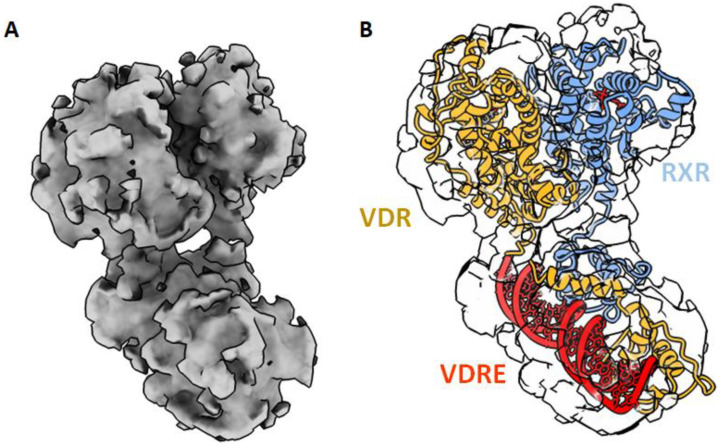
Cryo-EM structure of the VDR–RXR on DNA. (**A**) Electron density map shown as surface. (**B**) Fitted atomic model of the heterodimer in the EM map [59].

**Figure 5 nutrients-14-02847-f005:**
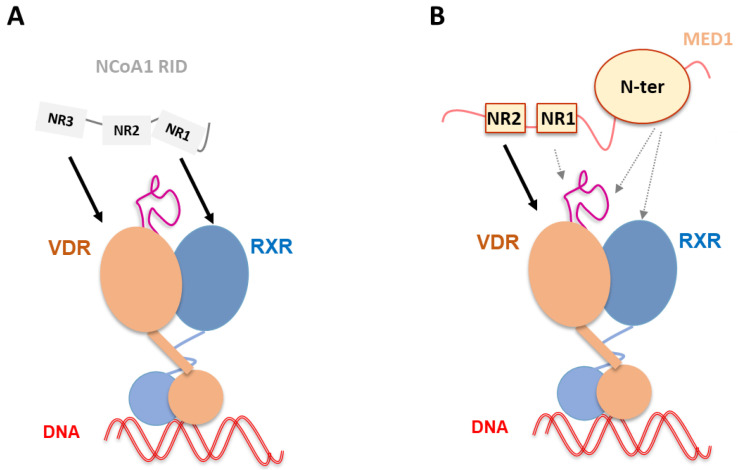
Schematic representation of NCoA1 and MED1 binding to liganded VDR–RXR–DNA complex. (**A**) NCoA1 RID binding to liganded VDR–RXR in complex with and without DNA [60]. The NCoA1 NR3 motif is associated to VDR and the NR1 to RXR. NCoA1 RID binding has been shown to stabilize not only VDR AF-2 but also RXR H3 and H10-H11. (**B**) MED1 (50–660) binding to liganded VDR–RXR–DNA complex [89]. Complex formation is primarily driven by strong ligand-dependent MED1 NR2 binding to the VDR AF-2, but other MED1 regions including NR1 and the structured N-terminal domain are involved in the interaction, as well as alternative sites of the receptors, including VDR insertion domain and RXR.

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
