# Peer review of "Vitamin D and Its Receptor from a Structural Perspective"

_nutrients, 2022, doi:10.3390/nu14142847_

Round 1

Reviewer 1 Report

This review provided an outline of existing VDR structural knowledge along with recent developments in understanding vitamin D's complex mechanism of action. The manuscript appears to be interesting. However, the author must address the following points before it may be considered for publication.

1. The abstract should be modified so that it explains why this topic is important to address and what approaches have been developed, used, and/or implemented to solve this type of problem;

2. The paper seems based on molecular docking, so, please add the importance of this approach to the review study;

3. The authors should revise the introduction section to include more information about the core topic, its importance, objectives, and research direction;

4. Literature Review has the chance to be further improved;

5. Through the review, what issues should be addressed? What is the current specific knowledge gap?;

Author Response

I'm pleased Reviewer 1 found the review interesting.

Responses to reviewer's comments:

  1. The abstract has been modified for clarification.
  2. The review described only experimental structural studies and no molecular docking.
  3. Introduction has been revised with the addition of few sentences on the aims of the review.
  4. Literature review could have been improved but all references related to structural studies of VDR complexes with 1,25D3 are cited.
  5. Concerning remaining questions, a sentence has been added in the introduction together with the perspectives paragraph.

Reviewer 2 Report

The manuscript (Review) by Natacha Rochel presents the overview of the current knowledge of VDR structures and explore the recent progress in understanding the complex mechanism of action of vitamin D. The information about ligand recognition and mechanism of action by structural analysis of the VDR complexes were well analysed. The data are interpreted appropriately and consistently throughout the manuscript. The review is clear and relevant for the field of study that’s why is appropriate for the special issue. The statements and conclusions are drawn coherent and supported by the listed references. I found the manuscript interesting although there are few common and specific comments to manuscript are presented.

Common comments

In the introduction, the author gives a description of the vitamin D3 and its main receptor, the sites of biosynthesis of this biologically active substance, reflects in a complex metabolic pathways in which the vitamin is involved and what pathologies can be. However, the author does not mention the aim of this review and the existence of the problem of understanding the complex mechanism of vitamin D activation. A few sentences should be added.

The figures images are appropriate. However, if the figures are not prepared by the author of the article, then references are required.

Specific comments

Line 23: … «in different species». Please, make a clarification - species of which group of organisms.

Line 43: It will be better to rename section 2 – Structure and mechanism of activation of the Vitamin D Receptor

Line 53-56: I guess, a links should be added.

Line 95: «4.1,25. D3 binding to VDR ligand binding domain». The numbers are merging. Please share.

Line 139-141: It is better to indicate generic or specific names also in Latin, as is generally accepted.

Line 173-174: The sentence should be rephrased. For example: The discussion of the structural implications of these missense mutations were presented in studies [48-49].

Line 183-184: The sentence should be rephrased.

Line 264-265: «Multiple biochemical and structural studies have mapped the interaction regions of 264 NRs with coactivator NR box». Several references to studies should be given.

Line 305: Remove repeated word (in)

Author Response

I'm pleased that Reviewer 2 found the review interesting.

Responses to Reviewer2 's comments:

Few sentences on the aims of the review have been included at the end of the introduction.

All figures are new but the references of thecorresponding studies are now included. 

Concerning the specific comments, all corrections were done.

Round 2

Reviewer 1 Report

All questions raised were responded and now the paper can be published.